# Poly-γ-glutamylation of biomolecules

Ghader Bashiri [1,2] ✉, Esther M. M. Bulloch[1,2], William R. Bramley[1], Madison Davidson [3], Stephanie M. Stuteley [1,2], Paul G. Young [1,2], Paul W. R. Harris[1,2], Muhammad S. H. Naqvi[1], Martin J. Middleditch[1], Michael Schmitz[4], Wei-Chen Chang [3], Edward N. Baker[1,2] & Christopher J. Squire [1,2] ✉

Poly-γ-glutamate tails are a distinctive feature of archaeal, bacterial, and eukaryotic cofactors, including the folates and $F_{420}$. Despite decades of research, key mechanistic questions remain as to how enzymes successively add glutamates to poly-γ-glutamate chains while maintaining cofactor specificity. Here, we show how poly-γ-glutamylation of folate and $F_{420}$ by folylpolyglutamate synthases and γ-glutamyl ligases, non-homologous enzymes, occurs via processive addition of ʟ-glutamate onto growing γ-glutamyl chain termini. We further reveal structural snapshots of the archaeal γ-glutamyl ligase (CofE) in action, crucially including a bulged-chain product that shows how the cofactor is retained while successive glutamates are added to the chain terminus. This bulging substrate model of processive poly-γ-glutamylation by terminal extension is arguably ubiquitous in such biopolymerisation reactions, including addition to folates, and demonstrates convergent evolution in diverse species from archaea to humans.

Poly-γ-glutamylation is a key feature of the folates (vitamin $B_9$) and cofactor $F_{420}$, and is considered as a general post-translational protein modification regulating diverse cellular functions[1]. Folates are essential cofactors central to one-carbon ($C_1$) metabolism[2], and are required for the biosynthesis of purines, pyrimidines and certain amino acids[3]. Poly-γ-glutamylation regulates folate homeostasis by promoting intracellular retention and modulates affinity with specific protein partners[4]. Cofactor $F_{420}$ is a flavin derivative pivotal in the primary and secondary metabolism of archaea and a wide range of bacteria, acting as a hydride carrier in diverse redox reactions[5]. The $F_{420}$ poly-γ-glutamate tail is proposed to control cellular redox homeostasis in bacteria by modulating affinity with $F_{420}$-dependent proteins[6]. The formation of a poly-γ-glutamate tail is catalyzed for the folates by folylpolyglutamate synthase[7] (FPGS), and for $F_{420}$ by the γ-glutamyl ligases FbiB[8,9] in bacteria and CofE[10,11] in archaea. These enzymes link together a variable number of ʟ-glutamate residues via their γ-carboxylates onto the cofactors (Fig. 1).

Poly-γ-glutamylation in both the folates and $F_{420}$ systems has long been presumed to follow an extension mechanism in which the incoming ʟ-glutamate is added to the terminus of the growing chain. Based on this broad mechanistic description, it is proposed in FPGS enzymes that each ʟ-glutamate addition proceeds via activation of the terminal γ-carboxylate on the growing substrate in an ATP-dependent manner to form an acyl-phosphate intermediate[12], and then nucleophilic attack by the incoming ʟ-glutamate to form a new amide bond. This mechanism is primarily inferred from the pioneering work of Banerjee et al. (1988) but remains to be definitively confirmed at a structural and mechanistic level. The human FPGS (hFPGS) system is even more cryptic since it processively catalyzes the addition of four ʟ-glutamates in each binding event[13]. This raises a long-standing question of how long poly-γ-glutamate tails of up to 14 residues are grown while the terminal glutamyl residue must always be positioned within the active site.

In considering the shortcomings of the current extension mechanism, we investigated two major questions. How do the enzymes maintain recognition of the core structures of the cofactors, providing substrate specificity, while simultaneously adding terminal

[1]School of Biological Sciences, The University of Auckland, Private Bag 92019, Auckland 1142, New Zealand. [2]Maurice Wilkins Center for Molecular Biodiscovery, The University of Auckland, Private Bag 92019, Auckland 1142, New Zealand. [3]Department of Chemistry, North Carolina State University, Raleigh, NC 27695, USA. [4]School of Chemical Sciences, The University of Auckland, Private Bag 92019, Auckland 1142, New Zealand. ✉e-mail: g.bashiri@auckland.ac.nz; c.squire@auckland.ac.nz

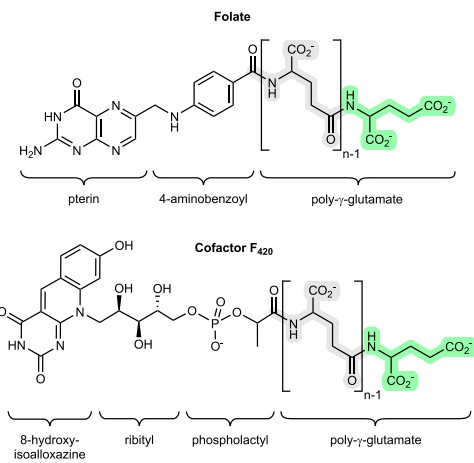

**Fig. 1 | Cofactor substrates of folylpolyglutamate synthase (folate) or γ-glutamyl ligase (F₄₂₀) bearing variable length poly-γ-glutamate tails.** The glutamyl residues in green indicate the terminus of the growing tail in both molecules.

glutamyl residues without indiscriminate off-target glutamylation occurring? And, how are the growing tails accommodated within the enzymes?

We have used chemical probe synthesis, mass spectrometry, nuclear magnetic resonance (NMR) spectroscopy, and X-ray crystallography to formulate a comprehensive model of poly-γ-glutamylation in representative FPGS, CofE, and FbiB enzymes from human, archaeal, and bacterial sources. Our results crucially reveal a bulge in the growing poly-γ-glutamate tail of our archaeal exemplar system, and lead to a mechanism that enables simultaneous recognition of the cofactor core structure, the cofactor terminal glutamyl residue, and the incoming L-glutamate co-substrate. This model provides a deeper molecular understanding of the elusive poly-γ-glutamylation process, suggesting that evolution has converged on a ubiquitous mechanism to grow long poly-γ-glutamate tails across the domains of life.

## Results

### An L-glutamate analogue terminates poly-γ-glutamylation

Studies of poly-γ-glutamylation are hampered by the lack of L-glutamate analogues that can act as enzyme substrates and useful molecular probes of enzymatic mechanism. We screened several L-glutamate analogues in enzymatic reactions using recombinant hFPGS[14], *M. tuberculosis* FPGS (*Mtb*-FPGS)[15], *M. tuberculosis* γ-glutamyl ligase (*Mtb*-FbiB)[8], and *Archaeoglobus fulgidus* γ-glutamyl ligase (*Af*-CofE)[11]. These systems represent eukaryotic, bacterial, and archaeal species, and comprise two non-homologous enzymes modifying either folate (FPGS) or F₄₂₀ (FbiB/CofE) substrates. Our rationale in these experiments was to identify an L-glutamate analogue that, once incorporated in a growing tail, prevents further reaction, effectively acting as a chain terminator or suicide substrate. To this end, we tested commercially-sourced analogues with γ-carboxylate substitutions, including 2-amino-L-4-sulfobutanoic acid (L-homocysteic acid), 4-azido-L-homoalanine, γ-methyl, t-butyl and benzyl esters of L-glutamic acid, and we synthesized 4-nitro-L-2-aminobutyric acid using published methods[16,17] (Supplementary Fig. 1 to 4).

LC-MS analysis of enzymatic reactions showed that all but one analogue, 4-nitro-L-2-aminobutyric acid, were either not substrates for poly-glutamylation in any system, or had their unique chemical features eliminated in reaching the final product, resulting in a canonical poly-glutamate tail, effectively invisible to mechanistic probing. The nitro-substituted probe, however, acts as a substrate across all enzymes and both cofactors, and generates non-reactive products after the addition of a single molecule. The diagnostic masses of these chain-terminated products were identified by high-resolution LC-MS

analysis (Fig. 2). The ubiquitous production of single-additive molecules, including in systems that otherwise produce longer-chained species, suggests that the analogue is added to the end of the substrate cofactors, terminating the poly-glutamylation reaction. These results provide direct experimental evidence supporting a terminal extension mechanism in either the folates or F₄₂₀.

### Isotopically labelled L-glutamate adds to the growing terminus of folic acid

We used ¹⁵N-labelled L-glutamate in a series of NMR experiments to further investigate terminal extension in an FPGS enzyme. An FPGS/folate system was chosen as an exemplar due to the intractability of this system to mechanistic elucidation over decades of experimentation. To provide the simplest set of interpretable spectra, the bacterial *Mtb*-FPGS enzyme and folate-2 substrate (containing two L-glutamyl residues) were chosen. This was informed by our LC-MS results detailing the addition of only a single L-glutamate molecule to afford folate-3 as the major product (Fig. 2). Coupled reactions comprising recombinant *Mtb*-FPGS[15] and dihydrofolate reductase from *Thermotoga maritima* (*Tm*-DHFR)[18] were used to increase turnover relative to *Mtb*-FPGS alone (Fig. 3a). *Tm*-DHFR reduces folates and dihydrofolates to tetrahydrofolates, with the latter being the preferred substrates for FPGS enzymes[14,19,20]. However, tetrahydrofolate is unstable, undergoing spontaneous oxidation at neutral pH (half-life <1 h)[21,22], and we anticipated all tetrahydrofolate molecules in NMR experiments to present predominantly as *N*-(4-aminobenzoyl) species (Fig. 3a).

Chemical shift assignments were first determined for folate-2 and folate-3 standards to guide the assignment of NMR product peaks. This was achieved through a series of 2D ¹H-¹H and ¹H-¹³C correlation experiments and aided by reported data for folate (Supplementary Table 1)[23,24]. Importantly, sequential assignments for the 1′, 2′ and 3′ glutamyl units of folate-3 were confirmed via strong through-space ¹H-¹H NOE correlations between each glutamyl amide and the γ-methylene group of the preceding glutamyl unit (see Fig. 3a for atom labels).

Polyglutamylation reaction samples contained folate-2 as substrate and ¹⁵N-enriched L-glutamate, and were analysed by recording 1D ¹H and 2D ¹H-¹⁵N HSQC spectra to monitor the glutamyl amide peaks of the substrates and reaction products. Prior to the addition of enzyme, two distinct amide peaks were present in the ¹H-¹⁵N HSQC spectra, corresponding to the 1′ ($\delta_H$ 7.92 ppm, $\delta_N$ 121.0 ppm) and 2′ ($\delta_H$ 7.87 ppm, $\delta_N$ 130.7 ppm) glutamyl units of folate-2 (Fig. 3b and Supplementary Fig. 5a). A two-hour pre-incubation with *Tm*-DHFR initiated the production of tetrahydrofolate-2, as confirmed by distinctive 1D ¹H peaks in the 3.5 to 3.0 ppm range corresponding to the tetrahydropterin ring (Supplementary Fig. 6)[18].

*Mtb*-FPGS was then added to the reaction mixture and spectra were recorded at approximately two-hour intervals (Supplementary Fig. 5c-h). At the first time point, folate-2 peaks were greatly reduced, and a single new amide peak emerged at a chemical shift of $\delta_H$ 7.83 ppm, $\delta_N$ 130.6 ppm (Fig. 3c and Supplementary Fig. 5c), increasing in intensity to a magnitude greater than the initial natural abundance peaks by 20 h (Fig. 3d and Supplementary Fig. 5h). From 22 h post-initiation, a longer ¹H-¹⁵N HSQC experiment was carried out to detect weaker ¹H-¹⁵N signals. This confirmed one intense and stable ¹⁵N-enriched peak ($\delta_H$ 7.83 ppm, $\delta_N$ 130.6 ppm) (Supplementary Fig. 7a and 7c), which was not present in a *Tm*-DHFR only control experiment (Supplementary Fig. 5b, 7b and 7c). Spontaneously oxidised tetrahydrofolate products, as anticipated (Fig. 3a), were confirmed in the reaction mixes as *N*-(4-aminobenzoyl)-(L-Glu)₂ and *N*-(4-aminobenzoyl)-(L-Glu)₃ enriched with ¹⁵N at a single site by mass spectrometry (Supplementary Fig. 8).

The chemical shifts for the ¹⁵N-enriched amide peak attributed to the *N*-(4-aminobenzoyl)-(L-Glu)₃ reaction product, are similar to that determined for the 3′ terminal glutamyl of folate-3 (Fig. 3e,

| Enzyme | Substrate mass | Predominant poly-glutamylated product mass | Nitro analogue mass | |
|---|---|---|---|---|
| | | | Observed | Expected |
| hFPGS | Folate-1 (441.1394) | Folate-6 (1086.3638) | Folate-2-NO$_2$ (571.1654) | Folate-2-NO$_2$ (571.1781) |
| | Folate-2 (570.1640) | Folate-6 (1086.3639) | Folate-3-NO$_2$ (700.1923) | Folate-3-NO$_2$ (700.2207) |
| | Folate-3 (699.2327) | Folate-6 (1086.3577) | Folate-4-NO$_2$ (829.2578) | Folate-4-NO$_2$ (829.2632) |
| *Mtb*-FPGS | Folate-1 (441.1394) | Folate-3 (699.2139) | Folate-2-NO$_2$ (571.1697) | Folate-2-NO$_2$ (571.1781) |
| | Folate-2 (570.1640) | Folate-3 (699.2187) | Folate-3-NO$_2$ (700.2023) | Folate-3-NO$_2$ (700.2207) |
| *Af*-CofE | F$_{420}$-1 (644.1389) | F$_{420}$-2 (773.1616) | F$_{420}$-2-NO$_2$ (774.1720) | F$_{420}$-2-NO$_2$ (774.1751) |
| *Mtb*-FbiB | F$_{420}$-1 (644.1389) | F$_{420}$-7 (1417.3367) | F$_{420}$-2-NO$_2$ (774.1662) | F$_{420}$-2-NO$_2$ (774.1751) |

**Fig. 2 | 4-Nitro-L-2-aminobutyric acid as a chain terminator of poly-γ-glutamylation in folates and F$_{420}$.** A poly-γ-glutamate tail is shown with variable L-glutamate numbers and terminated by a non-reactive nitro functional group. Calculated masses of nitro-terminated folate and F$_{420}$ molecules are provided for comparison to experimental results – the central number indicates the total residue count in the tail including the chain terminator residue. The masses take into account functional group protonation within the mass spectrometer operating in positive ion mode and are provided to 4 decimal places to match the precision of the high-resolution LC-MS experiment. The table of data provides the identity and experimentally measured masses of substrate, predominant polyglutamylated product, and nitro-terminated product. Enzymes systems investigated are human FPGS (hFPGS), *Mycobacterium tuberculosis* FPGS (*Mtb*-FPGS), *Archaeoglobus fulgidus* CofE (*Af*-CofE), and *Mycobacterium tuberculosis* FbiB (*Mtb*-FbiB).

Supplementary Table 1). To confirm the site of the [15]N incorporation, 2D [1]H-[1]H TOCSY and NOESY experiments were carried out on the reaction sample, both with [15]N-coupling and with [15]N-decoupling applied (Fig. 3f to 3i and Supplementary Table 1). The [1]H NMR peak of the [15]N-enriched amide is split by 91 Hz in the [15]N-coupled experiments, in accordance with reported one-bond [15]N-[1]H coupling constants for amides[25]. Comparison of the [15]N-decoupled and [15]N-coupled 2D [1]H-[1]H TOCSY spectra reveals through-bond correlations between the [15]N-enriched amide and aliphatic protons with chemical shifts of 4.09, 2.00, 1.86, and 2.19 ppm (Fig. 3h). These chemical shifts align with those of the 3′α, 3′β$_1$, 3′β$_2$ and 3′ γ protons, respectively, of the folate-3 standard (Fig. 3g and Supplementary Table 1). Furthermore, the [15]N-decoupled NOESY spectrum shows a through-space correlation with a proton at a distinctive 2.26 ppm chemical shift (Fig. 3i) matching that of the 2′γ protons of folate-3 (Fig. 3g and Supplementary Table 1). We conclude that *Mtb*-FPGS incorporates [15]N-glutamate at the terminus of the polyglutamyl chain of tetrahydrafolate-3.

**Structural snapshots of archaeal γ-glutamyl ligase CofE in action prove poly-glutamyl "bulging"**

The archaeal γ-glutamyl ligase CofE was chosen as the most tractable exemplar system to identify structural features required for poly-γ-glutamylation. The γ-glutamyl ligase enzymes *Mtb*-FbiB[8] and *Af*-CofE[11] catalyze F$_{420}$ poly-γ-glutamate tail formation in bacteria (*Mycobacterium tuberculosis*) and archaea (*Archaeoglobus fulgidus*) respectively. FbiB is a two-domain protein that requires both domains to add multiple L-glutamates (up to 14) to F$_{420}$[8]. The simpler CofE system comprises a single-domain protein homologous to the N-terminal domain of FbiB, but generating predominantly F$_{420}$-2 (i.e., with two L-glutamyl residues) but also larger F$_{420}$-3 and F$_{420}$-4 products in our experiments (Supplementary Fig. 9).

We determined a series of *Af*-CofE crystal structures with a systematic combination of reaction ligands, including GTP/Mn$^{2+}$, GDP/Mn$^{2+}$/F$_{420}$-1, GTP/Mn$^{2+}$/F$_{420}$-2, and GTP/Mn$^{2+}$/L-glutamate, all to high resolutions of 1.3–1.7 Å (Supplementary Table 2). Previous CofE structures have defined its dimeric structure and metal ion and GTP binding details[11]. Our structures now reveal unambiguous binding models for all ligands involved in the reaction, including F$_{420}$-1 (a substrate), F$_{420}$-2 (the predominant product) and incoming L-glutamate (Fig. 4a and Supplementary Fig. 10). Each of the ligands binds within the complex active site of the enzyme, located in the dimer interface and spanning a distance of -38 Å (Supplementary Fig. 11a). A long domain-swapped β-hairpin loop wraps over one side of the elongated active site forming critical contacts with the F$_{420}$ cofactors. The various ligands form extensive, complementary interactions with the protein, which include coordination to two divalent metal ions (Mn$^{2+}$), to water molecules bound to these metal ions, and to a presumed monovalent cation (Supplementary Fig. 11b and 11c). Cofactor binding induces only minor conformational change with root-mean-square deviation (rmsd) values of 0.108-0.171 Å (203-230 Cα coordinates) resulting when structures are overlaid.

These *Af*-CofE structures in complex with GTP/Mn$^{2+}$ and GDP/Mn$^{2+}$/F$_{420}$-1 now show, in exquisite detail, how the F$_{420}$-1 substrate sits poised for GTP-dependent activation in preparation for binding of a second glutamate. The F$_{420}$-1 isoalloxazine chromophore binds into a pocket surrounded by an array of nonpolar side chains at its back face, and an extended polypeptide section (G153-C155) and dimer partner arginine (R234) at its front face; the cationic side chain of R234 π-

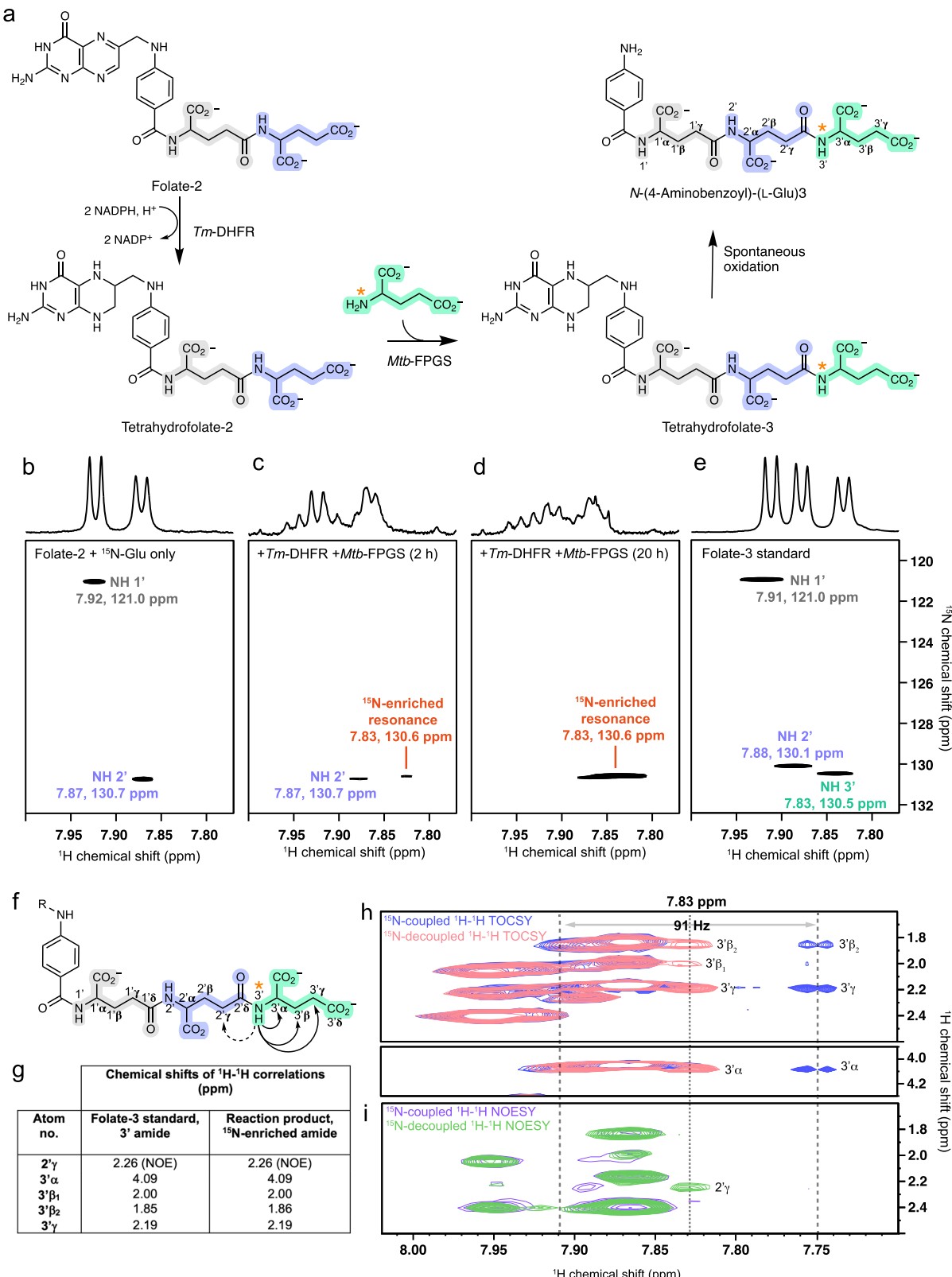

stacks above the pyrimidine ring and the F92 aromatic ring stacks behind the central pyridine ring (Supplementary Fig. 12a). The 8-hydroxyl of the isoalloxazine ring hydrogen bonds via bridging waters to the protein backbone and additionally back to an $F_{420}-1$ phosphate oxygen. These interactions, alongside further hydrogen bond interactions between the ribityl and phospholactyl moieties and

the protein, firmly lock the isoalloxazine ring into this conserved site (Supplementary Fig. 12b).

In the $F_{420}-1$ complex, the amide carbonyl oxygen hydrogen bonds with two water molecules, each coordinated to a different manganese ion. In the presence of a GTP γ-phosphate, as modelled in Fig. 4b, the amide carbonyl oxygen would be located 3.3 Å away from

**Fig. 3 | *Mtb*-FPGS-dependent glutamylation of folate-2 with [15]N-glutamate monitored by NMR spectroscopy. a** Reaction scheme for *Mtb*-FPGS/*Tm*-DHFR coupled conversion of folate-2 to tetrahydrofolate-3 and subsequent oxidation to *N*-(4-aminobenzoyl)-(L-Glu)₃. The orange asterisk indicates the [15]N-enriched site. Panels **b** to **e** show the amide region of 1D [1]H and 2D [1]H-[15]N HSQC spectra for samples as follows. **b** Reaction sample containing folate-2 and [15]N-glutamate prior to addition of enzyme. **c** Reaction sample following a 2 h preincubation with *Tm*-DHFR, followed by incubation with both *Tm*-DHFR and *Mtb*-FPGS for 2 h and **d** 20 h. **e** Folate-3 standard. Panels **f** to **i** show the results of [1]H-[1]H correlation experiments confirming addition of [15]N-glutamate at the terminus of the *Mtb*-FPGS/*Tm*-DHFR-

catalyzed reaction product. **f** Arrows indicate through-bond (solid line) and through-space (dashed line) correlations observed between the [15]N-labelled amide and other [1]H nuclei in the polyglutamate chain, based on folate-3 assignments. **g** Comparison of the chemical shifts of [1]H-[1]H correlations observed for the [15]N-enriched amide and the 3′ glutamyl amide of folate-3. **h** Overlay of [15]N-coupled (blue), and [15]N-decoupled (pink) [1]H-[1]H TOCSY experiments for the [15]N-enriched amide highlighting correlations with the 3′α, 3′β, and 3′γ protons. **i** Overlay of [15]N-coupled (purple), and [15]N-decoupled (green) [1]H-[1]H NOESY experiments for the [15]N-enriched amide highlighting a correlation with the 2′γ proton.

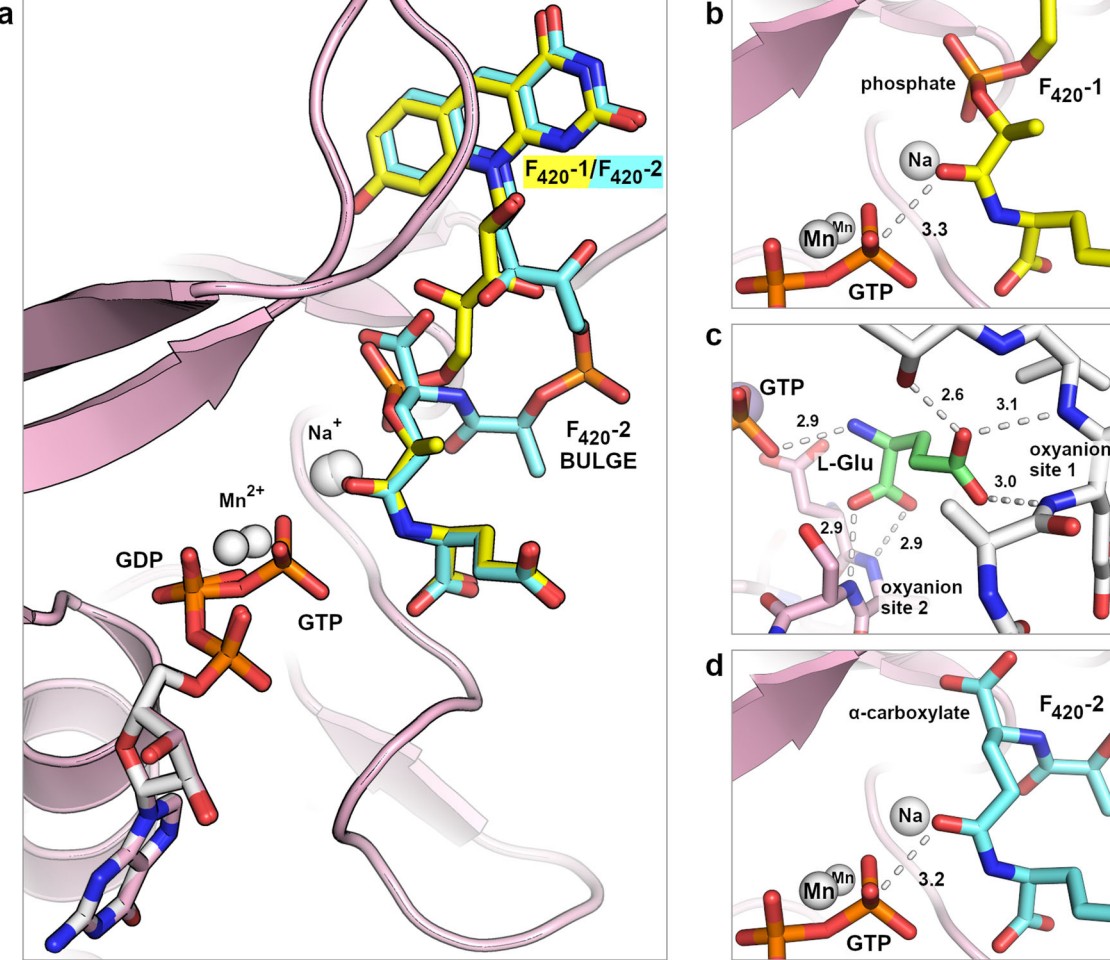

**Fig. 4 | High-resolution structures of reaction ligands in the active site of *Archaeoglobus fulgidus* γ-glutamyl ligase CofE. a** An overlay of F₄₂₀-1 (yellow) and F₄₂₀-2 (cyan) molecules highlighting the bulge in the tail of F₄₂₀-2. GTP/GDP (pink stick model), manganese and sodium cations (white spheres) are located adjacent to the terminal glutamyl binding site. **b** The F₄₂₀-1 phospholactyl moiety binds in a secondary glutamate binding pocket and its carbonyl closely approaches the γ-phosphate atom of a modelled GTP. **c** Free L-glutamate binding in the primary

glutamate pocket. Extensive hydrogen bonding between the co-substrate and two oxyanion-like holes via α- and γ-carboxylates locks the ligand in place. **d** The first, basal glutamyl of F₄₂₀-2 occupies the secondary glutamate pocket and projects the carbonyl oxygen of the following *cis*-amide bond toward the GTP γ-phosphate atom. Polar interactions are shown as dashed white lines with distances labelled in Å units.

the phosphorous atom. This amide, linking the F₄₂₀−1 phospholactyl to the glutamyl residue, is observed in a high energy *cis* configuration that projects the glutamyl into an oxyanion-like binding pocket. Confirming the significance of this site, in our GTP/Mn²⁺/L-glutamate structure this same pocket is occupied by a free L-glutamate locked in between two oxyanion-like holes via its α- and γ-carboxylates (Fig. 4c). With both F₄₂₀-1 glutamyl and the incoming L-glutamate sharing near identical ligand-protein interactions, we propose that this dimer interface site is a conserved, high-affinity or primary glutamate binding pocket in CofE and its homologue FbiB.

The GTP/Mn²⁺/F₄₂₀-2 structure affords a view of how a growing poly-glutamyl chain (two glutamyl residues in this case) is accommodated in the enzyme active site. While interactions between the isoalloxazine chromophore and its protein and solvent environment closely resemble those of the F₄₂₀-1 structure, F₄₂₀-2 binds with a 180° torsional rotation immediately past the first ribityl hydroxyl, creating a bulge that projects the remainder of the ribityl moiety and the phosphate group out of the binding site toward solvent space (Fig. 4a). At the same time, the second ribityl hydroxyl forms a strong internal hydrogen bond to the α-carboxylate of the first glutamyl, effectively

**Fig. 5 | Schematic of the proposed poly-γ-glutamylation mechanism in CofE and its homologue FbiB.** Isoalloxazine and primary glutamate binding sites are depicted as bold, dotted lines, with glutamate or glutamyl residues colored green or blue by their order of addition. In STEP 1, $F_{420}$-0 effects nucleophilic attack on the GTP γ-phosphate to form an acyl intermediate. The phosphate abstracts a proton from the free glutamate amino group and is eliminated from GTP. STEP 2 shows an additional proton abstraction, nucleophilic attack of the glutamate on the acyl phosphate, and the consequent and concerted loss of phosphate as $H_2PO_4^-$. STEP 3 depicts the crystal structure model of $F_{420}$-1 binding. STEP 4 and STEP 5 show a second cycle of phosphorylation/activation, proton abstraction, nucleophilic attack, and elimination of $H_2PO_4^-$. STEP 6 depicts the crystal structure model of $F_{420}$-2. STEPS 4–6 highlight the bulge in the growing poly-glutamyl chain. The schematic has been simplified for clarity and multiple binding, bond making, and bond breaking events have been combined – this does not imply a specific order of binding nor concerted reactions.

linking together the base of the bulging conformation. This carboxylate also connects back to the 8-hydroxyl of the isoalloxazine via a bridging water molecule, further stabilising this unique conformation of $F_{420}$-2 (Supplementary Fig. 13). The phosphate at the midpoint of the bulge is further locked in place by water mediated hydrogen bonds to the protein. This bulge places the first, basal, glutamyl residue of $F_{420}$-2 into a secondary glutamate binding pocket, equivalent to the binding pocket of the phospholactyl moiety in the $F_{420}$-1 structure. The amide bond joining the second glutamyl to the first has a *cis* configuration, orienting its carbonyl oxygen towards the GTP γ-phosphate (Fig. 4d), specifically, 3.2 Å away from the phosphorous atom. Once again, the terminal glutamyl is directed into the conserved, primary glutamate pocket by the *cis* configuration of the adjacent amide bond.

These crystal structure snapshots provide the mechanistic details of a multi-step enzymatic mechanism involving primary and secondary glutamate binding interactions, phosphoryl transfer/substrate activation, phosphate elimination coupled with glutamate nucleophilic attack, and substrate bulging. The implications of these crystal structures and a detailed enzymatic mechanism are discussed below.

## Discussion

Our study has answered a key and long-unresolved question concerning this crucial bio-polymerization reaction, specifically how the poly-γ-glutamyl tails of the folates and $F_{420}$ grow, often to considerable lengths, while maintaining the terminal glutamyl within the machinery of the active site. Our experiments, reported here, using 4-nitro-L-2-aminobutyric acid and $^{15}N$-labelled L-glutamate as chemical probes of the poly-γ-glutamylation of two different systems, folate and the cofactor $F_{420}$, deliver definitive experimental evidence that poly-γ-glutamylation does proceed by the addition of L-glutamate molecules to the terminus of the growing chain, as had been hypothesized. This applies across the domains of life and in non-homologous enzymes.

The mechanism requires several enzymatic features that are revealed in the series of CofE crystal structures presented here. These capture snapshot states of this γ-glutamyl ligase in action, including the elusive proof of bulging-out of the growing chain to allow the terminal glutamate to remain in the active site (Fig. 4a). How does poly-γ-glutamylation in this system work mechanistically?

Our crystal structures enable visualization of all the ligands involved in the γ-glutamyl ligase reaction in various combinations, allowing us to formulate a detailed mechanism of poly-γ-glutamylation in CofE (Fig. 5). The starting point in our catalytic mechanism shows the binding of $F_{420}$-0, which we infer from the $F_{420}$-1 model, and places the pholactate carboxylate immediately adjacent to the γ-phosphate of GTP (Fig. 5, Step 1), as expected. The GTP binds to divalent metal ions, polarising the terminal phosphate group and facilitating nucleophilic attack by the $F_{420}$-0 carboxylate oxygen on the phosphorus, forming a substrate adduct (acyl intermediate). The terminal phosphate in accordance with its expected $pK_a$ and in the absence of any other catalytic base, abstracts a proton from the L-glutamate located in the primary glutamate binding pocket (Fig. 4c). The phosphoryl oxygens rearrange through inversion and torsional rotation, and the phosphoryl abstracts another proton from the L-glutamate (Fig. 5, Step 2). The amino group of the glutamate is

directed toward the acyl-phosphate intermediate, orchestrating nucleophilic attack with the concerted elimination of dihydrogen phosphate ($H_2PO_4^-$). The resulting $F_{420}$-1 molecule retains its single glutamyl in the primary glutamate binding site (Fig. 5, Step 3). The binding of new GTP and L-glutamate molecules displaces the $F_{420}$-1 glutamyl towards the cofactor chromophore to now occupy the secondary binding site previously filled by the phospholactyl moiety, producing a bulge in the growing tail (Fig. 5, Step 4).

This concurrent binding of the displaced $F_{420}$-1 glutamyl in a secondary site, with retention of isoalloxazine chromophore binding, necessitates bulge formation, and most importantly, places the terminal γ-carboxylate adjacent to the γ-phosphate of GTP for the enzymatic cycle to repeat. A second round of phosphoryl transfer, acyl-intermediate formation, proton abstraction, and L-glutamate nucleophilic attack coupled with phosphate elimination affords the $F_{420}$-2 product (Fig. 5, Steps 4–6). We propose that throughout chain extension, this mechanism is processive with the cofactor substrate residing in the enzyme active site until sufficient glutamyl chain length is attained to drive product release.

Our proposed mechanism bears strong similarity to that of the non-homologous enzyme MurD in its addition of a single D-glutamate to a UDP-N-acetylmuramoyl-L-alanine substrate in bacterial peptidoglycan synthesis[26]. The MurD mechanism comprises phosphoryl transfer from ATP to the terminal carboxylate of the substrate, nucleophilic attack of a D-glutamate amino group to effect ligation, and the elimination of phosphate. Substrate and ATP-binding is scaffolded on two divalent metal centers and the glutamate occupies an oxyanion-like binding site. The ligands (ATP, substrate, and D-glutamate) are arrayed spatially in a similar way to the molecules in CofE catalysis (Supplementary Fig. 14). Most critically, the second enzyme focus of our own studies, FPGS, is a MurD homologue that differs most obviously to MurD in its addition of multiple L-glutamate residues to its folate substrates. While several FPGS crystal structures are available[15,27–29], ligand binding information is scarce, with ATP (or non-hydrolysable analogues) the most prevalent ligand in these structures. A single folate analogue, phosphorylated dihydropteroate (DHPP) has, however, been trapped in a crystal structure of FolC[30], a bifunctional dihydrofolate synthase/folylpolyglutamate synthase that catalyzes the addition of a glutamate to dihydropteroate and the successive addition of multiple glutamates to tetrahydrofolate. By overlaying multiple liganded structures of MurD, FolC, and FPGS, we can make a composite model of FPGS ligand binding focusing on the missing element of L-glutamate binding (Supplementary Fig. 14).

Our model suggests that L-glutamate binds into a site equivalent to our CofE primary glutamate site, placing the amino group adjacent to the phosphorylated substrate to effect nucleophilic attack on the substrate. The reactive carbonyl of the folate substrate and its activating phosphate are coordinated to a divalent metal – metal binding of the carbonyl both holds it in an appropriate geometry and polarizes the C = O bond to facilitate the first glutamylation event. Following this glutamylation, we propose that another L-glutamate enters the putative primary binding site as the previously added glutamate shuffles into the site previously occupied by the phosphorylated carbonyl of the substrate and binds to the divalent metal center. The FPGS active site displays a path to solvent space to accommodate substrate bulging as we propose for FbiB.

Questions remain about the mechanism of chain length determination in the enzymes we have studied. Neither CofE/FbiB or FPGS systems possess a molecular ruler by which to measure chain length, and theoretically, any glutamyl chain length might be accommodated by further extension or bulging-out into the aqueous environment surrounding the protein. We propose that chain length is determined through a thermodynamics-based phenomenon, in which a tension exists between active site binding and glutamyl interactions with the protein surface and/or bulk water molecules. In essence, at longer

chain lengths, the high entropy of the poly-γ-glutamyl chain sampling multiple conformations in solution and the enthalpic contribution of forming strong solvent hydrogen bonds, pulls the cofactor molecule from the active site. Our proposed mechanism may also offer an explanation for the unusual processivity observed in hFPGS, in which it catalyzes the addition of four L-glutamate residues in a given binding event[13]. This proposal awaits a robust computational analysis including molecular dynamics simulation to argue compellingly.

Finally, we consider the question of whether our proposed poly-γ-glutamylation mechanism can be extrapolated beyond the CofE/FbiB and FPGS comparison and inform us about the likelihood of a universal mechanism. The many commonalities between the FPGS and CofE/FbiB systems suggests a broadly similar mechanism might be shared. Our mass spectrometry (Fig. 2) and NMR spectroscopy (Fig. 3 and Supplementary Fig. 5–7) experiments with both $F_{420}$ and folate substrates, together with the experimental evidence for an acyl-phosphate intermediate in FPGS catalysis[12], unambiguously prove that γ-glutamyl ligases and folylpolyglutamate synthases catalyze the addition of L-glutamate molecules at the terminus of the growing tail. It is plausible, and likely, that the folate chromophore in FPGS, like $F_{420}$ in CofE, and the terminal glutamyl residue of the growing poly-γ-glutamate chain in both systems, must always be held simultaneously in the enzyme active site for terminal extension to occur. By analogy with our CofE conclusions, alternation between primary and secondary glutamate binding pockets in the FPGS active site, could enable catalysis to proceed with a bulging γ-glutamyl chain. Given these parallels, it seems conceivable that nature will have convergently evolved the same strategy to effect poly-γ-glutamylation in other disparate enzymes across the domains of life.

## Methods

### Protein expression and purification

**hFPGS**. The open reading frame (ORF) encoding hFPGS[14] was cloned into pFastBacHTa using EcoRI and XbaI restriction sites. The construct, possessing an N-terminal His$_6$-tag, was expressed in Sf9 insect cells (Protein Expression Facility, The University of Queensland, Australia). The protein was purified by immobilized metal affinity chromatography (IMAC) using Talon resin (Clontech, Takara Bio.) and size-exclusion chromatography (SEC) on a Superdex 200 10/30 column (GE Healthcare), in a base buffer of 20 mM Tris-HCl pH 7.5, 50 mM KCl, 50 mM NaCl, 2% glycerol, 2 mM β-mercaptoethanol.

**Mtb-FPGS**. The ORF encoding FPGS[15] of *M. tuberculosis* H37Rv was cloned into pYUB28b[31] with the resulting construct expressing His$_6$-tagged protein in *M. smegmatis* mc$^2$4517 cells[32]. The cells were grown in a fermenter (BioFlo®415, New Brunswick Scientific) for 4 days and the protein purified using Ni-NTA and SEC steps, as described for hFPGS, and using a base buffer of 20 mM HEPES pH 8.0, 100 mM KCl, 1 mM MgCl$_2$, 5% glycerol, 2 mM β-mercaptoethanol.

**Af-CofE**. The codon-optimized ORF encoding *A. fulgidus* CofE[11] was synthesized and cloned (GenScript) into pProEX-HTb for expression in *E. coli* BL21 (DE3) LOBSTR cells[33]. The expression cells were lysed in 20 mM HEPES pH 7.5, 500 mM NaCl, 20 mM imidazole and 1 mM β-mercaptoethanol, and the supernatant loaded onto a Ni-NTA column for washing with 100 mL lysis buffer containing 2 M urea and 2 M KBr to separate the co-purified flavins from the protein[34]. After on-column removal of the N-terminal His$_6$-tag using rTEV protease[35], the colorless protein was further purified on a Superdex 75 SEC column using 20 mM HEPES pH 7.5, 150 mM NaCl, 5% glycerol, 1 mM β-mercaptoethanol.

**Mtb-FbiB**. The wild-type *Mtb*-FbiB was expressed and purified as previously described[8]. The protein was purified in 20 mM Tris-HCl pH 7.5, 150 mM NaCl, 1 mM β-mercaptoethanol.

***Tm*-DHFR.** The ORF encoding dihydrofolate reductase from *Thermotoga maritima* was synthesized and cloned (GenScript) into pProEX-HTb for expression in *E. coli* BL21 (DE3) LOBSTR cells[33]. The expression cells were lysed in Tris·HCl pH 7.0, 100 mM NaCl, 50 mM KCl, 20 mM imidazole, 5% glycerol, and the protein purified using TALON resin (Clontech, Takara Bio.) followed by SEC purification on a Superdex 75 column.

### $F_{420}$-1 and $F_{420}$-2 preparation
$F_{420}$ was purified from *Mycobacterium smegmatis* cells as previously described[31]. The purified $F_{420}$ was treated with carboxypeptidase G (*Pseudomonas* sp., Sigma) to partially remove the poly-γ-glutamate tail, affording $F_{420}$-1 and $F_{420}$-2. The two $F_{420}$ species were then purified on a Waters (Milford, MA, USA) Breeze HPLC system using a C18 semi-preparative reverse phase HPLC column with a water/acetonitrile solvent mixture.

### General chemical synthesis procedures
The chemical shift values are reported in δ values (parts per million, ppm) relative to the standard chemical shift for the hydrogen residue peak and carbon-13 peak in the deuterated solvent, $CDCl_3$, or $D_2O$[36]. The coupling constant (*J*) values are expressed in hertz (Hz). Thin-layer chromatography (TLC) was performed on silica gel plates. Compounds on TLC were visualized by illumination under UV light (254 nm), dipped into $KMnO_4$, or 10% phosphomolybdic acid solution followed by charring on a hot plate. Silica gel (230–400 mesh) was used for flash column chromatography. Evaporations were carried out under reduced pressure (water aspirator or vacuum pump) with the bath temperature below 50 °C unless specified otherwise. Materials obtained from commercial suppliers were used directly without further purification.

### Preparation of (2 *S*)-2-amino-4-nitrobutanoic acid
**(2 *S*)-4-Nitro-2-[bis[(1,1-dimethylethoxy)carbonyl]amino]-butanoic acid methyl ester.** (*S*)-methyl-2-bis(ᵗbutoxycarbonyl)amino-4-iodobutanoate was prepared in 20–50 mmol scale from L-aspartate by literature procedures (Supplementary Fig. 2)[16,17]. To a stirred solution of methyl (2 *S*)-2-[bis[(1,1-dimethylethoxy)carbonyl]-4-iodobutanoate (1.2 g, 2.81 mmol) in dimethylformamide (DMF, ~5 mL) was added sodium nitrite (387 mg, 5.62 mmol) at room temperature. After stirring for 2 h, the reaction was concentrated under reduced pressure and subjected to silica gel flash column chromatography using an eluent of hexanes/ethyl acetate (4/1, v/v) to yield (2 *S*)-4-nitro-2-[bis[(1,1-dimethylethoxy)carbonyl]amino]-butanoic acid methyl ester in 50% yield. $^1H$ NMR (500 MHz, $CDCl_3$): δ 4.98 (dd, 1H, *J* = 8.5 and 5.5 Hz, 1H), 4.49 (m, 2H), 3.73 (s, 3H), 2.86 (m, 1H), 2.53 (m, 1H), 1.49 (s, 18H). $^{13}C$ NMR (125 MHz, $CDCl_3$): δ 170.0, 151.7, 84.0, 72.2, 55.4, 52.5, 27.9, 27.7 (Supplementary Fig. 3).

**(2 *S*)-2-Amino-4-nitrobutanoic acid.** To a solution of (2 *S*)-4-nitro-2-[bis[(1,1-dimethylethoxy)carbonyl]-methyl ester (462 mg, 1.28 mmol) was dissolved in 1,4-dioxane/$H_2O$ (1/1, total of 8 mL), followed by addition of lithium hydroxide (LiOH) (36 mg, 1.53 mmol) at room temperature. The reaction was monitored via TLC using ethyl acetate/methanol (4/1) as an eluent. The reaction was concentrated under reduced pressure and subjected to silica gel flash column chromatography using eluent of DCM/methanol (4/1) to yield (2 *S*)-2-[bis[(1,1-dimethylethoxy)carbonyl]amino-4-nitro-butanoic acid.

To a solution of (2 *S*)-2-[bis[(1,1-dimethylethoxy)carbonyl]amino-4-nitro-butanoic acid (77 mg, 0.22 mmol) in DCM (~5 mL) was added trifluoroacetic acid (TFA) (16 μL, 2.2 mmol) at room temperature. After full consumption of the starting material, the reaction was concentrated under reduced pressure to yield (2 *S*)-2-amino-4-azido-butanoic acid as the TFA salt in 70% yield over two steps (Supplementary Fig. 2). $^1H$ NMR (500 MHz, $D_2O$): δ 3.88 (t, *J* = 7.0 Hz, 1H), 2.55

(m, 4H). $^{13}C$ NMR (125 MHz, $D_2O$): δ 170.8, 70.8, 50.2, 26.7 (Supplementary Fig. 4).

### LC-MS analyses
FPGS activity assays[37] were performed using pteroylmono-γ-L-glutamic acid (folate-1; Sigma), pteroyldi-γ-L-glutamic acid (folate-2; Schircks Laboratories) or pteroyltri-γ-L-glutamic acid (folate-3; Schircks Laboratories) in 20 mM Tris·HCl pH 8.5, 50 mM NaCl, 25 mM KCl, 5 mM ATP, 10 mM $MgCl_2$, 10 mM $NaHCO_3$ and 1 mM tris(2-carboxyethyl)phosphine (TCEP). The folates were used at 0.5 mM and either natural or analogues of L-glutamic acid were added at 5 mM concentration. FbiB/CofE assays[8,11] were carried out in 20 mM Tris·HCl pH 8.5, 50 mM KCl/NaCl, 2 mM $MgCl_2$/$MnCl_2$, 2 mM GTP, 50 μM $F_{420}$-1 and either natural or analogues of L-glutamic acid at 2 mM. All reactions were incubated at 37 °C, followed by injection onto a C18 trap cartridge (Thermo Scientific) for desalting prior to chromatographic separation on a 0.3 × 100 mm 3.5 μm Zorbax 300SB C18 Stablebond column (Agilent Technologies, Santa Clara, CA, USA) equilibrated with acetonitrile/water. The column eluate was ionised in the electrospray source of a QSTAR-XL Quadrupole Time-of-Flight mass spectrometer (Applied Biosystems, Foster City, CA, USA). For IDA (Information Dependent Analysis) analyses, a TOF-MS scan from 400–1600 m/z was performed, followed by three rounds of MS/MS on the most intense singly or doubly charged precursors in each cycle. For targeted work, defined Product Ion Scans were created to isolate and fragment specific ions of interest with various collision energies (20–80 kV). Both positive and negative modes of ionisation were used as appropriate.

### LC-MS/MS analyses
The digests were acidified and injected onto a 0.3 × 10 mm trap column packed with Reprosil C18 media (Dr. Maisch) and desalted for 10 min at 10 μl/min before being separated on a 0.075 × 200 mm picofrit column (New Objective) packed in-house with 3 μm Reprosil C18-AQ media. The following gradient was applied at 300 nl/min using a NanoLC 425 UPLC system (Eksigent): 0 min 5% B; 16 min 40% B; 18 min 98% B; 20 min 98% B; 20.5 min 5% B; 30 min 5% B, where A was 0.1% formic acid in water, and B was 0.1% formic acid in acetonitrile.

The picofrit spray was directed into a TripleTOF 6600 Quadrupole-Time-of-Flight mass spectrometer (Sciex) for Information Dependent Analysis (IDA), comprising a TOF-MS scan from 300-2000 m/z for 200 ms, followed by 75 ms MS/MS scans on the 15 most abundant singly and doubly-charged species (m/z 50–1200), using Rolling Collision Energy and High Sensitivity mode, for a total cycle time of ~1.4 s. The mass spectrometer and UPLC system were under the control of the Analyst TF 1.8 software package (Sciex).

### NMR experiments
Both *Mtb*-FPGS and *Tm*-DHFR were purified as described above and buffer exchanged to 20 mM Tris ($D_{11}$, 98%, Cambridge Isotopes) pH 7.0, 100 mM NaCl, 50 mM KCl prior to the NMR spectroscopy. NMR spectroscopy experiments were run on a Bruker AV 600 spectrometer equipped with a triple-resonance pulsed-field gradient cryoprobe. NMR spectra were processed using TopSpin 4.1.4 (Bruker). Water suppression for $^1H$ NMR experiments was by excitation sculpting[38]. A sensitivity enhanced pulse program (BEST-HSQC) was employed for $^1H$-$^{15}N$ HSQC experiments[39]. Non-uniform sampling (10% sparse) was used for $^1H$-$^{13}C$ HMBC experiments[40,41]. Chemical shifts for folate-2 and folate-3 were from standard samples at concentrations of 1.5 to 3.6 mM in 10% v/v $D_2O$, directly adjusted to pH 7.0 or 8.0 with NaOH, and with 50 μM sodium trimethylsilylpropanesulfonate (DSS) as an internal reference (δ 0 ppm). Chemical shifts were assigned for the polyglutamate chains of folate-2 and folate-3 using a combination of the following experiments: 1D $^1H$, 1D $^{13}C$, $^1H$-$^{15}N$ HSQC, $^1H$-$^1H$ COSY, $^1H$-$^1H$ TOCSY, $^1H$-$^1H$ NOESY, $^1H$-$^{13}C$ HSQC, and $^1H$-$^{13}C$ HMBC.

Experiments probing the site of $^{15}$N-glutamate incorporation in folate-2 by *Mtb*-FPGS were run at 37 °C. Samples contained 20 mM Tris (D$_{11}$, 98%, Cambridge Isotopes) pH 7.0, 0.1 mM DTT, 4 mM MgCl$_2$, 10 mM NaHCO$_3$, 2 mM ATP, 2 mM NADPH, 1.5 mM $^{15}$N-glutamate (98%, Cambridge Isotopes), 1 mM folate-2, 10% v/v D$_2$O, and 0.25 mM DSS as a chemical shift and intensity reference. For the *Tm*-DHFR/*Mtb*-FPGS reaction, *Tm*-DHFR (8 μM) was added first, followed by a 2 h incubation period. Then *Mtb*-FPGS (20 μM) was added and the incubation was continued, with 1D $^{1}$H and 2D $^{1}$H-$^{15}$N HSQC (32 scan) experiments collected at 110 min intervals for the first 20 h. The control sample was incubated over an equivalent time period with *Tm*-DHFR (8 μM) only. Higher sensitivity $^{1}$H and $^{1}$H-$^{15}$N HSQC experiments (380 scans) were run on both the reaction and control samples after incubation for 22 h. The final *Tm*-DHFR/*Mtb*-FPGS reaction sample was further analysed with $^{1}$H-$^{1}$H COSY, $^{1}$H-$^{1}$H TOCSY ($^{15}$N-coupled and $^{15}$N decoupled), $^{1}$H-$^{1}$H NOESY ($^{15}$N-coupled and $^{15}$N decoupled) and $^{1}$H-$^{13}$C HMBC experiments. In addition, the final composition of the *Tm*-DHFR/*Mtb*-FPGS NMR reaction sample was analysed via LC-MS using the methods detailed in the section above.

### *Af*-CofE crystallization and structure determination

The *Af*-CofE protein was prepared for crystallization by removing the co-purified flavins[11] with extensive washing using 2 M urea and 2 M KBr supplemented buffer[34]. This step was crucial to obtain co-crystals with the reaction ligands. The resulting colorless protein was co-crystallized with ligand combinations of 2 mM GTP/5 mM Mn$^{2+}$, or 2 mM GTP/5 mM Mn$^{2+}$/1 mM F$_{420}$-1, or 2 mM GTP/5 mM Mn$^{2+}$/1 mM F$_{420}$-2, or 2 mM GTP/5 mM Mn$^{2+}$/1 mM L-glutamate using a sitting drop vapor diffusion method and crystallization buffer comprising 0.8 M ammonium sulfate, 0.1 M citrate pH 4.5. Data collection statistics are summarized in Supplementary Table 2. All data sets were indexed and processed using XDS[42], and scaled with AIMLESS[43] from the CCP4 program suite[44]. The structures of *Af*-CofE in complex with different ligands were solved by molecular replacement using PHASER[45] with the previous *Af*-CofE structure (PDB code 2PHN[11]) as a search model. The structure was refined by cycles of automatic model building by phenix.autobuild[46] and manual building using COOT[47], followed by refinement using REFMAC5[48]. Full refinement statistics are shown in Supplementary Table 2. Attempts to solve crystal structures containing F$_{420}$-1/glutamate or F$_{420}$-2/glutamate via co-crystallization or soaking experiments were unsuccessful.

### FPGS ligand modelling

To produce an FPGS ligand model, MurD, FolC, and FPGS crystal structure coordinates (2UAG, 4UAG, 8DP2, 1W78, and 2VOR) were downloaded from the Protein Data Bank[49]. The structures were overlaid using COOT and appropriate ligands appended to the FPGS structure as indicated by structural similarity. An L-glutamate was modelled into place by comparison with succinate or product binding locations and intermolecular contacts in PDB structures 8DP2 and 4UAG, respectively. The modelled glutamate was adjusted manually in COOT to fit its γ-carboxylate into a hypothetical oxyanion-like hole formed by the hydroxyl of serine 473 and amide NH groups of valines 474 and 475, making appropriate polar contacts also between its α-carboxylate and arginine 412.

### Figure preparation

Molecular structures and schematic drawings were drawn using ChemDraw Professional version 21.0.0.28 (PerkinElmer Informatics). All crystal structure figures were produced using the PyMOL Molecular Graphics System, Version 4.6 Schrödinger, LLC.

### Reporting summary

Further information on research design is available in the Nature Portfolio Reporting Summary linked to this article.

## Data availability

The data that support the findings of this study are available within the paper, its supplementary information files, or at the following repositories with the specified accession codes. Protein crystal structures reported in this study have been deposited in the Protein Data Bank (PDB; https://www.wwpdb.org/) under accession codes 7ULD, 7ULE, 7ULF, and 8G8P. The coordinate data used in molecular modelling and illustrated in Supplementary Fig. 14 were obtained from the Protein Data Bank under accession codes 2UAG, 4UAG, 8DP2, 1W78, and 2VOR.

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

## Acknowledgements

We acknowledge Prof. Mark Distefano (Department of Chemistry, University of Minnesota) for suggesting the use of a nitro L-glutamate analogue in our experiments. We thank Dr. Erica Prentice (University of Waikato, New Zealand) for data collection at the Australian Synchrotron. GB is supported by a Sir Charles Hercus Fellowship through the Health Research Council of New Zealand (17/058). This research was undertaken in part using the MX2 beamline at the Australian Synchrotron, part of ANSTO, and made use of the Australian Cancer Research Foundation (ACRF) detector. Access to the Australian Synchrotron was supported by the New Zealand Synchrotron Group Ltd.

## Author contributions

G.B. conceived project, designed experiments, performed experiments, analyzed results and co-wrote the manuscript. E.M.M.B. conceived project, designed experiments, performed experiments, and analyzed results with the assistance of M.S., and co-wrote the manuscript. W.R.B., M.D., S.S. and M.S.H.N. performed experiments and analyzed results with input from M.J.M., P.G.Y. and P.R.W.H. W.C.C. designed experiments, analyzed results and co-wrote the manuscript. E.N.B. conceived project, analyzed results and co-wrote the manuscript. C.J.S. conceived project, designed experiments, conducted molecular modelling experiments, analyzed results, and co-wrote the manuscript. All authors provided feedback on the manuscript.

## Competing interests

The authors declare no competing interests.

## Additional information

**Supplementary information** The online version contains Supplementary Material available at https://doi.org/10.1038/s41467-024-45632-1.

