## [Peer Review File · Nature Communications]

REVIEWER COMMENTS

Reviewer #1 (Remarks to the Author):

Title: Poly- γ -glutamylation of biomolecules

Polyglutamylation of cofactors such as folate and F420 are key post-translational modifications required for folate retention and affinity modification in F420-dependent enzymes, respectively. The reaction is catalyzed by folylpolyglutamate synthase (FPGS) in the case of folates, and by γ -glutamyl ligases FbiB and CofE in the case of F420. In both cases the mechanism involves the ATP-dependent ligation of a free L-glutamate to a growing polyglutamate tail but details concerning the reaction mechanisms are limited. For example, is the incoming glutamate inserted into the growing tail at the beginning where it is joined to the folate or F420, or is it added to the end the glutamate tail? If it is the latter, the terminal glutamate must always be positioned in the active site, raising the question of where is the growing tail accommodated? Moreover, given that enzymes from different species can give rise to products of different polyglutamate tail lengths, what triggers the release of the product at a particular length? This article describes some elegant biochemical studies involving 4-nitro-L-2-aminobutyric acid and ¹⁵N-labelled L-glutamate as probes of the γ -glutamylation reaction which show that the reaction does indeed proceed via addition to the terminus of the glutamate tail in both the folate and F420 systems. The structure determination of several substrate- and cofactor-bound complexes of archaeal CofE from *Archaeoglobus fulgidus* are also reported, which suggest that the growing tail bulges out from the active site in such a way that the terminal glutamate is always presented within the active site in an orientation conducive to further γ -glutamylation.

The authors present a well-written and thoughtful analysis of the mechanism of action of the *A. fulgidus* CofE enzyme. I found the manuscript enjoyable to read and very easy to follow. The authors come from a group who have made major contributions in the folate polyglutamylation field for many years. The experimental work and subsequent analyses were carried out appropriately, and I deem the results to be a major step forward in the understanding of this very important post-translational process. I highly recommend this article for publication in *Nature Communications*.

I have no major issues and just a few minor comments which need attention.

Minor comments:

Discussion

I think the data presented here does indicate that the glutamate residues in both the folate and F420 systems are indeed added sequentially to the end of the tail, and it is gratifying to see this conclusion. This has always seemed to be the most rational mechanism to me, given that an insertion reaction would require the breaking of an amide bond and the formation of two new amide bonds, which would be highly energetically unfavored, the latter most likely requiring two GTP or ATP molecules. Moreover, the "terminal ligation" mechanism also makes logical sense given that the bacterial cell wall amide bond ligases (MurC, MurD, MurE and MurF) do an almost identical reaction with tertiary structures highly similar to FPGS. Although the four enzymes catalyze one ligation reaction each, the ligations are sequential at the terminal carboxylate moiety. Perhaps the authors should comment on this?

The proposed mechanism for CofE is very plausible. I have a couple of minor points:

Line 296. The authors state that “An incoming free L-glutamate then binds the primary glutamate binding pocket” following formation of the acyl intermediate. It is also very likely, is it not, that the free glutamate could be bound prior to GTP or F420 binding? I do not think the structures presented here can equivocally state which substrates/cofactors bind first, second or third. The presence of the glu/GTP complex suggests that F420-0 binding does not necessarily need to be the first event and that prior to Step 2, the F420-0, GTP and glutamate could enter in any order. However it is clear that the F420 with the growing tail must remain attached to the enzyme for the subsequent steps with new GTP and glutamate entering.

Line 300: “The binding of a new GTP molecule and another free L-glutamate displaces the F420-1 glutamyl ...”. Did the authors try to crystallize the F420-1/glutamate or F420-2/glutamate complexes? Presumably these would show the pre-ligation state where the terminal glutamate has transitioned into the secondary binding site. The latter complex, were it able to be obtained, might also start to answer the question as to how the growing bulge is accommodated. If the authors did attempt to make these complexes and were unable, perhaps this could be mentioned in the methods. The complexes were all made by co-crystallization, correct? The authors could try to generate a F420-1/glutamate complex by soaking into pre-formed apo-CofE crystals if they feel that this may be worthwhile.

In steps 2 and 5, is there a nearby general base (or even a water molecule) which can accept the proton from the α -amino group on the free glutamate?

The presence of the positively charged patches on the molecular surface adjacent to the active site are intriguing and do suggest that the “bulging-out” model has credibility. The elongated positive patch in FbiB, which makes longer glutamate tails, is also consistent with this model. As to the mechanism of release of the complex, I agree that it is likely to be thermodynamically-driven.

Have the authors considered using molecular docking to see whether they can first reproduce the F420-2 complex, and then move on to longer derivatives? This could be done on the FbiB model to see if the growing bulge does interact with the elongated positive patch. In addition, although this is probably outside the scope of the current study, the authors should also consider applying molecular dynamics simulations to any docked complexes they can generate to get an understanding of the stability, flexibility and dynamics of the complexes, and what factors may be involved in retaining or releasing the elongated products. This could be applied to the FPGS and Mur systems as well, to obtain a more universal understanding of the formation of elongated amino acid tails on biological molecules.

Methods

This is not a criticism but I am curious: what criteria were used to determine the resolution cutoff for the different data sets? There doesn't appear to be consistency in Table S2 with respect to the R_{pim} , I/σ and $CC_{1/2}$ values. For example, based on these statistics, the data for the F420-2/GTP structure looks like it may extend to slightly higher resolution if cutoffs similar to those used for the other structures were employed (ie an I/σ closer to 1.5 and a $CC_{1/2}$ closer to 0.5). The lower average protein B values for this complex relative to F420-1/GDP might also suggest this. Was there any major conformational changes observed between the complexes? What are the rmsds between the complexes? The authors should provide this information, perhaps in the Supplementary Material.

Reviewer #2 (Remarks to the Author):

The paper by Bashiri et al investigates the distinctive feature of poly- γ -glutamate tails in various cofactors across archaeal, bacterial, and eukaryotic domains, such as folates and F420. Despite extensive research, questions persist regarding the mechanistic details of how enzymes sequentially add glutamates to these chains while maintaining cofactor specificity. The study demonstrates that polyglutamate synthases and γ -glutamyl ligases, non-homologous enzymes, achieve poly- γ -glutamylation of folate and F420 through the processive addition of L-glutamate onto growing γ -glutamyl chain termini. The authors present structural insights into the mechanism of the archaeal γ -glutamyl ligase (CofE) in action, revealing a "bulged-chain" product that illustrates how the cofactor is retained during successive glutamate additions to the chain terminus. The proposed bulging substrate model suggests a potentially universal mechanism in biopolymerisation reactions, suggesting convergent evolution across diverse species.

Paper makes a significant contribution and will be of general interest to researchers across a wide range of fields. The proposed mechanism for chain elongation is rational and entirely feasible, and in line with other known biochemical reactions. Furthermore, it is backed up by the crystal structure that have been determined.

Figure 3, panel E has misannotated chemical shifts for the NH 2'. The labels do not match chemical shift of the peak of the subsequent NMR data table.

I have concerns with the crystal structure and map presented in figure S9, panels A and B. It appears from the figure that there are clusters of magnesium ions coordinated together. This would not be possible and not supported by the observed density either. Presumably this might be a mistake with the figure, and these are actually a Mn²⁺ with coordination from 3 water molecules?

For the hypothesis proposed for controlling chain length. The authors could easily test this by introducing mutations in the positively charged groove proposed to accommodate and stabilise the growing poly-glutamate bulge.

If the chain length is thermodynamically controlled, have authors considered measuring binding constant by SPR or ITC and determining these thermodynamic parameters?

Reviewer #1

Polyglutamylation of cofactors such as folate and F420 are key post-translational modifications required for folate retention and affinity modification in F420-dependent enzymes, respectively. The reaction is catalyzed by polyglutamate synthase (FPGS) in the case of folates, and by γ -glutamyl ligases FbiB and CofE in the case of F420. In both cases the mechanism involves the ATP-dependent ligation of a free L-glutamate to a growing polyglutamate tail but details concerning the reaction mechanisms are limited. For example, is the incoming glutamate inserted into the growing tail at the beginning where it is joined to the folate or F420, or is it added to the end the glutamate tail? If it is the latter, the terminal glutamate must always be positioned in the active site, raising the question of where is the growing tail accommodated? Moreover, given that enzymes from different species can give rise to products of different polyglutamate tail lengths, what triggers the release of the product at a particular length? This article describes some elegant biochemical studies involving 4-nitro-L-2-aminobutyric acid and ^{15}N -labelled L-glutamate as probes of the γ -glutamyl reaction which show that the reaction does indeed proceed via addition to the terminus of the glutamate tail in both the folate and F420 systems. The structure determination of several substrate- and cofactor-bound complexes of archaeal CofE from *Archaeoglobus fulgidus* are also reported, which suggest that the growing tail bulges out from the active site in such a way that the terminal glutamate is always presented within the active site in an orientation conducive to further γ -glutamyl reaction.

The authors present a well-written and thoughtful analysis of the mechanism of action of the *A. fulgidus* CofE enzyme. I found the manuscript enjoyable to read and very easy to follow. The authors come from a group who have made major contributions in the folate polyglutamyl reaction field for many years. The experimental work and subsequent analyses were carried out appropriately, and I deem the results to be a major step forward in the understanding of this very important post-translational process. I highly recommend this article for publication in *Nature Communications*.

I have no major issues and just a few minor comments which need attention.

Minor comments:

Discussion

I think the data presented here does indicate that the glutamate residues in both the folate and F420 systems are indeed added sequentially to the end of the tail, and it is gratifying to see this conclusion. This has always seemed to be the most rational mechanism to me, given that an insertion reaction would require the breaking of an amide bond and the formation of two new amide bonds, which would be highly energetically unfavored, the latter most likely requiring two GTP or ATP molecules. Moreover, the "terminal ligation" mechanism also makes logical sense given that the bacterial cell wall amide bond ligases (MurC, MurD, MurE and MurF) do an almost identical reaction with tertiary structures highly similar to FPGS. Although the four enzymes catalyze one ligation reaction each, the ligations are sequential at the terminal carboxylate moiety. Perhaps the authors should comment on this?

We thank the referee for directing us to look more closely at the Mur enzymes. These enzymes catalyse a single L-Glu (or other amino acid or amino acid-like) addition, while our systems of interest can add multiple L-Glu residues. We agree, the mechanism proposed is

very similar with the putative phosphoryl transfer and nucleophilic attack of the free amino functionality of the incoming L-Glu residue.

Further prompted by the MurD analysis of L-glutamate addition, we have overlaid several MurD, FolC (adding L-Glu to dihydropteroate and tetrahydrofolates), and FPGS (adding L-Glu to tetrahydrofolates) structures. Structural similarities suggests that FPGS follows a similar mechanism to both MurD and CofE/FbiB with the substrate glutamyl chain bulging as it grows.

Our FPGS model features:

- 1) A path to solvent for a growing folate glutamyl chain.
- 2) An oxyanion-like primary glutamate site that is similar to that found in CofE/FbiB (specifically a serine-OH/amide-NH/amide-NH oxyanion like pocket).

This new analysis strengthens an argument for a universal mechanism common to multiple enzymes across the domains of life and we have revised the main article text to reflect this.

The proposed mechanism for CofE is very plausible. I have a couple of minor points:

Line 296. The authors state that “An incoming free L-glutamate then binds the primary glutamate binding pocket” following formation of the acyl intermediate. It is also very likely, is it not, that the free glutamate could be bound prior to GTP or F420 binding? I do not think the structures presented here can equivocally state which substrates/cofactors bind first, second or third. The presence of the glu/GTP complex suggests that F420-O binding does not necessarily need to be the first event and that prior to Step 2, the F420-O, GTP and glutamate could enter in any order. However it is clear that the F420 with the growing tail must remain attached to the enzyme for the subsequent steps with new GTP and glutamate entering.

We agree with the reviewer that our structures do not provide unequivocal evidence of the order of binding. It is possible that the L-Glu could bind prior to GTP or F420-O binding – and as noted – the Glu/GTP structure supports this. In an improved mechanism figure (Figure 5 in the revised manuscript), we do not differentiate between cofactor and glutamate binding order and present them together in each panel. We have changed the text description of the mechanism to provide clarity on the potential binding order, that is, that we cannot draw any conclusions.

Line 300: “The binding of a new GTP molecule and another free L-glutamate displaces the F420-1 glutamyl ...”. Did the authors try to crystallize the F420-1/glutamate or F420-2/glutamate complexes? Presumably these would show the pre-ligation state where the terminal glutamate has transitioned into the secondary binding site. The latter complex, were it able to be obtained, might also start to answer the question as to how the growing bulge is accommodated. If the authors did attempt to make these complexes and were unable, perhaps this could be mentioned in the methods. The complexes were all made by co-crystallization, correct? The authors could try to generate a F420-1/glutamate complex by soaking into pre-formed apo-CofE crystals if they feel that this may be worthwhile.

Attempts were made to elucidate all potential structures including those combinations suggested by the reviewer. Both co-crystallisation and soaking experiments were pursued over a number of months and multiple data collection trips to the synchrotron. We were unable to trap such complexes and we cannot offer any suggestion why this is the case – attempts to trap complexes for crystallography are often unpredictable in outcome. We have added a note in the methods at the end of the crystallography section that reads “Attempts to solve crystal structures containing F420-1/Glu or F420-2/Glu via co-crystallisation or soaking experiments were unsuccessful.”

In steps 2 and 5, is there a nearby general base (or even a water molecule) which can accept the proton from the α -amino group on the free glutamate?

This is an excellent suggestion from the reviewer and an oversight on our part not to include such analysis and discussion. In our newly formulated mechanism, we propose that phosphate acts as the catalytic base. Our crystal structures do not show any amino acid functionality, nor a water molecule, appropriately arranged to act in this way. However, the terminal phosphate of GTP is arranged in the ideal geometry and distance to function as the catalytic base. Furthermore, the mechanism (new Figure 5) we now propose including the catalytic base functionality of the phosphate and the elimination of H_2PO_4^- , is consistent with the pK_a values of the phosphate species involved [$\text{H}_2\text{PO}_4^- \rightleftharpoons \text{HPO}_4^{2-} + \text{H}^+$, $\text{pK}_{a2} = 7.20$].

The presence of the positively charged patches on the molecular surface adjacent to the active site are intriguing and do suggest that the “bulging-out” model has credibility. The elongated positive patch in FbiB, which makes longer glutamate tails, is also consistent with this model. As to the mechanism of release of the complex, I agree that it is likely to be thermodynamically-driven.

Have the authors considered using molecular docking to see whether they can first reproduce the F420-2 complex, and then move on to longer derivatives? This could be done on the FbiB model to see if the growing bulge does interact with the elongated positive patch. In addition, although this is probably outside the scope of the current study, the authors should also consider applying molecular dynamics simulations to any docked complexes they can generate to get an understanding of the stability, flexibility and dynamics of the complexes, and what factors may be involved in retaining or releasing the elongated products. This could be applied to the FPGS and Mur systems as well, to obtain a more universal understanding of the formation of elongated amino acid tails on biological molecules.

The reviewer suggestions are very useful and in fact outline our current plans for a follow-on paper. We have completed docking experiments with FbiB and an F_{420-7} cofactor. The results suggest that the growing polyglutamyl chain is long enough to reach deeply into the positively charged groove of FbiB but also that at least 50% of the time it extends out into solvent space and does not interact with the protein surface. Other binding poses sampled include interaction with another positively charged patch near the active site groove. The mixture of binding modes aligns with the review comments suggesting that some more robust computational modelling including molecular dynamics (MD) will be required to probe our thermodynamic hypothesis. Our intention is to complete comprehensive

computational studies and publish as a follow up to the current article focused exclusively on the thermodynamics of chain length determination.

Given the preliminary docking analysis, we have rewritten the section relating to the thermodynamic discussion of chain length control and removed the electrostatic surface imagery that we believe is no longer relevant.

Methods

This is not a criticism but I am curious: what criteria were used to determine the resolution cutoff for the different data sets? There doesn't appear to be consistency in Table S2 with respect to the R_{pim} , I/σ and $CC\frac{1}{2}$ values. For example, based on these statistics, the data for the F420-2/GTP structure looks like it may extend to slightly higher resolution if cutoffs similar to those used for the other structures were employed (ie an I/σ closer to 1.5 and a $CC\frac{1}{2}$ closer to 0.5). The lower average protein B values for this complex relative to F420-1/GDP might also suggest this.

We agree with the reviewer, that crystals no doubt diffracted a little higher in resolution. The current data cut-off was imposed by the data collection geometry, that is, the crystal to detector distance could have been better chosen to allow collection to higher resolution – the dataset is processed to its maximum limit.

Was there any major conformational changes observed between the complexes? What are the rmsds between the complexes? The authors should provide this information, perhaps in the Supplementary Material.

We now present rmsd information in the main text. When overlaid on the CofE complex with GTP, the rmsd values for the other three structures range between 0.108 and 0.171 Å covering between 203 and 230 C α atoms – the structures all overlay very closely.

Reviewer #2

The paper by Bashiri et al investigates the distinctive feature of poly- γ -glutamate tails in various cofactors across archaeal, bacterial, and eukaryotic domains, such as folates and F420. Despite extensive research, questions persist regarding the mechanistic details of how enzymes sequentially add glutamates to these chains while maintaining cofactor specificity. The study demonstrates that folylpolyglutamate synthases and γ -glutamyl ligases, non-homologous enzymes, achieve poly- γ -glutamylation of folate and F420 through the processive addition of L-glutamate onto growing γ -glutamyl chain termini. The authors present structural insights into the mechanism of the archaeal γ -glutamyl ligase (CofE) in action, revealing a "bulged-chain" product that illustrates how the cofactor is retained during successive glutamate additions to the chain terminus. The proposed bulging substrate model suggests a potentially universal mechanism in biopolymerisation reactions, suggesting convergent evolution across diverse species.

Paper makes a significant contribution and will be of general interest to researchers across a wide range of fields. The proposed mechanism for chain elongation is rational and entirely feasible, and in line with other known biochemical reactions. Furthermore, it is backed up by

the crystal structure that have been determined.

Figure 3, panel E has misannotated chemical shifts for the NH 2'. The labels do not match chemical shift of the peak of the subsequent NMR data table.

Figure 3 has been updated.

I have concerns with the crystal structure and map presented in figure S9, panels A and B. It appears from the figure that there are clusters of magnesium ions coordinated together. This would not be possible and not supported by the observed density either. Presumably this might be a mistake with the figure, and these are actually a Mn²⁺ with coordination from 3 water molecules?

Many thanks to the reviewer for noticing this oversight. In the different panels of the image, we have drawn coordinated water molecules in two different ways, and this has caused the confusion. This was not intentional and as the reviewer suggests, the Mn²⁺ is coordinated with water molecules – it is not a cluster of metals. The water-coordinated metals have been re-labelled in the image to make this explicit.

For the hypothesis proposed for controlling chain length. The authors could easily test this by introducing mutations in the positively charged groove proposed to accommodate and stabilise the growing poly-glutamate bulge. If the chain length is thermodynamically controlled, have authors considered measuring binding constant by SPR or ITC and determining these thermodynamic parameters?

These are excellent suggestions and we have considered such binding experiments in the past. However, F420 derivatives including F420-1 and F420-2 are not commercially available and the yields we can produce in the laboratory are not sufficient to conduct the proposed binding experiments. The extensive time involved and associated cost of producing the cofactor samples are primary factors given we no longer have funding for this work. As we have proposed above, we believe the best path forward towards understanding this system and the potential for thermodynamic chain length control, is to undertake robust docking and MD experiments. Nonetheless, MD work in this complex system would take some months to complete and is beyond the scope of the current communication article with its focus on terminal addition with bulging, and other mechanistic features not related to chain length determination. We have modified our language in relation to the discussion of thermodynamics – our intention in the current article was to add this for interest and an indication of “what comes next”. Our intention is to do this computational work and to publish as a follow-up to the current communication.

REVIEWERS' COMMENTS

Reviewer #1 (Remarks to the Author):

I would like to thank the authors for their thoughtful response to my questions and comments. Upon reading through the response letter and the revised manuscript, I am very pleased to confirm that all my concerns have been adequately met. I unreservedly recommend that this article is published in Nature Communications without any additional revisions.

Reviewer #2 (Remarks to the Author):

I am satisfied that the authors have fully addressed my initial concerns and I would like to recommend the article for publication in Nature Communications.